

# Integrating phylogeographic and ecological niche approaches to delimitating cryptic lineages in the blue–green damselfish (*Chromis viridis*)

Shang Yin Vanson Liu[1], Mao-Ning Tuanmu[2], Rita Rachmawati[3,4], Gusti Ngurah Mahardika[5] and Paul H. Barber[4]

[1] Department of Marine Biotechnology and Resources, National Sun Yat-Sen University, Kaohsiung, Taiwan
[2] Biodiversity Research Center, Academia Sinica, Taipei, Taiwan
[3] Center for Fisheries Research, Ministry of Marine Affairs and Fisheries, Jakarta, Indonesia
[4] Department of Ecology and Evolutionary Biology, University of California, Los Angeles, CA, USA
[5] The Indonesian Biodiversity Research Centre, Udayana University, Bali, Indonesia

## ABSTRACT

Species delimitation is challenging in sibling species/cryptic lineages because of the absence of clear diagnostic traits. However, integration of different approaches such as phylogeography and ecological niche comparison offers one potential approach to tease apart recently diverged lineages. In this study, we estimate the ecological niche divergence among lineages in *Chromis viridis* in a broad-scale phylogeographic framework to test whether the combination of these two approaches can effectively distinguish recently diverged lineages. Results from Cytb and Rag2 analyses identified two cryptic lineages (*C. viridis A* and *C. viridis B*) that diverged ~3 Myr ago. Estimates of ecological niche divergence with 11 environmental parameters across the broad geographic range of these lineages showed overlapping ecological niches and niche conservatism. However, regardless of the incongruence between genetic and ecological niche divergence, the substantial genetic divergence between the two clades of *C. viridis* in both mtDNA and nuclear loci strong suggest that they are cryptic taxa.

## INTRODUCTION

Comparative phylogeography examines patterns of congruence in phylogenetic breaks across species distribution. It is built on the assumption that the processes driving lineage divergence, speciation, and the evolution of biodiversity mainly involves geographical, historical, and environmental factors that favor isolation and limit gene flow between populations (*Avise, 2000*; *Rocha, Craig & Bowen, 2007*).

In the marine realm, phylogeographic studies frequently identify cryptic species, morphologically indistinguishable groups that have differences in neutral genetic markers that are equal or greater than those observed between species with diagnostic

Corresponding author
Shang Yin Vanson Liu,
syvliu@mail.nsysu.edu.tw

morphological traits (*Knowlton, 1993*). In some cases (*Drew & Barber, 2009*; *Drew, Allen & Erdmann, 2010*; *Hubert et al., 2012*; *Liu et al., 2012*) such cryptic lineages are subsequently described as new species (*Allen & Drew, 2012*; *Liu et al., 2012*; *Allen, Erdmann & Kurniasih, 2015*), indicating that such cryptic lineages represent overlooked biodiversity.

Systematists use different characteristics to differentiate species, and the variation in these characteristics can arise at different times and rates during the process of lineage diversification (*De Queiroz, 2007*). However, species delimitation is particularly difficult in adaptive or recent radiations, when nascent species boundaries and their defining characteristics can be unclear (*Puebla et al., 2007*; *Wagner et al., 2013*; *Victor, 2015*). In such cases, integration of different approaches can increase the ability to detect recently separated lineages (*Leaché et al., 2009*) and can provide stronger evidence for lineage separation when the results are concordant (*Liu et al., 2012*; *Allen & Erdmann, 2012*; *Larkin et al., 2016*).

While integrating morphological diversification and genetic divergence is relatively common in addressing recent lineage diversification (*Allen & Drew, 2012*; *Allen & Erdmann, 2012*), ecological diversification (i.e., niche divergence) is seldom used to differentiate cryptic lineages/species in the marine realm. Niche diversification is a foundation of speciation (*Pyron & Burbrink, 2009*). Therefore, if niche divergence leads to lineage differentiation in the process of populations adapting to new environments (*Wiens, 2004*; *Wiens & Graham, 2005*), then a niche partitioning should be expected between lineages, particularly among recently diverged overlapping cryptic lineages identified in many phylogeographic studies.

To date, only a handful of studies have found niche divergence between cryptic marine taxa. For example, studies on deep-sea octocorals showed that niche partitioning between lineages is associated with the depth (*Quattrini et al., 2013*; *Quattrini, Gómez & Cordes, 2017*) and the marine cyanobacteria (genus *Prochlorococcus*) exhibits niche partitioning in associated with geography and environmental conditions. The absence of niche differentiation in the study of cryptic marine taxa is partially due to the lack of a centralized high-resolution spatial data representing both benthic and pelagic marine environments (*Sbrocco & Barber, 2013*), resources that are now available.

*Chromis viridis* is widely distributed Indo-Pacific coral reef fish. Both juveniles and adults associate with *Acropora* corals where they school and feed on zooplankton in the water column above coral heads (*Frédérich et al., 2009*). Using phylogenetic analyses based on mtDNA variation, *Froukh & Kochzius (2008)* found three cryptic lineages in *C. viridis* across four Indo-Pacific localities, and *Messmer et al. (2012)* documented the presence of additional cryptic lineages in the Great Barrier Reef. However, it is unclear how geographical and ecological processes contribute to this nascent diversification.

To better understand the processes driving lineage diversification in *C. viridis*, we conducted a broad scale phylogeographic study of *C. viridis* across its distribution range. We then estimated ecological niche divergence between lineages in *C. viridis* in this

**Table 1 Sampling locations and diversity indices based on *Cytb* sequences (911 bp) in 15 populations of *Chromis viridis* from the Indo-Pacific.**

| Abbreviations | Location | Cytb | | | | Rag2 | | | |
|---|---|---|---|---|---|---|---|---|---|
| | | N | nh | h ± SD | π ± SD | N | nh | h ± SD | π ± SD |
| RED | Eliat, Israel | 10 | 8 | 0.956 ± 0.059 | 0.004 ± 0.002 | 6 | 6 | 1.000 ± 0.096 | 0.003 ± 0.002 |
| MD | Toliara, Madagascar | 9 | 6 | 0.889 ± 0.091 | 0.007 ± 0.004 | 9 | 9 | 1.000 ± 0.052 | 0.005 ± 0.003 |
| AMED | Amed, Bali, Indonesia | 19 | 14 | 0.953 ± 0.036 | 0.007 ± 0.004 | 18 | 17 | 0.994 ± 0.021 | 0.004 ± 0.002 |
| LEM | Nusa Lembongan, Bali, Indonesia | 2 | 2 | 1.000 ± 0.500 | 0.003 ± 0.004 | 2 | 2 | 1.000 ± 0.500 | 0.006 ± 0.006 |
| KOMO | Komodo, Indonesia | 18 | 11 | 0.882 ± 0.064 | 0.014 ± 0.007 | 18 | 18 | 1.000 ± 0.019 | 0.005 ± 0.003 |
| XS | Xisha, China | 6 | 6 | 1.000 ± 0.096 | 0.006 ± 0.004 | 8 | 8 | 1.000 ± 0.063 | 0.004 ± 0.003 |
| DS | Dongsha, Taiwan | 15 | 11 | 0.933 ± 0.054 | 0.005 ± 0.003 | 15 | 15 | 1.000 ± 0.024 | 0.005 ± 0.003 |
| NU | NPP III Inlet, Taiwan | 31 | 24 | 0.972 ± 0.020 | 0.005 ± 0.003 | 23 | 23 | 1.000 ± 0.013 | 0.004 ± 0.002 |
| SK | SesokoIsland, Japan | 11 | 9 | 0.946 ± 0.066 | 0.005 ± 0.003 | 11 | 11 | 1.000 ± 0.034 | 0.005 ± 0.003 |
| SP | Saipan, USA | 2 | 1 | 0.000 ± 0.000 | 0.000 ± 0.000 | 2 | 2 | 1.000 ± 0.500 | 0.000 ± 0.000 |
| MI | Marshall Island, R.O. Marshall Islands | 20 | 15 | 0.942 ± 0.043 | 0.005 ± 0.003 | 13 | 13 | 1.000 ± 0.030 | 0.004 ± 0.003 |
| NC | New Caledonia, French | 17 | 16 | 0.993 ± 0.023 | 0.006 ± 0.003 | 15 | 15 | 1.000 ± 0.024 | 0.004 ± 0.002 |
| LZ | Lizard Island, Australia | 35 | 26 | 0.968 ± 0.020 | 0.039 ± 0.019 | 35 | 35 | 0.998 ± 0.007 | 0.009 ± 0.005 |
| FJ | Fiji | 5 | 5 | 1.000 ± 0.127 | 0.054 ± 0.033 | 4 | 4 | 1.000 ± 0.177 | 0.015 ± 0.011 |
| FP | Moorea Island, French Polynesia | 48 | 21 | 0.785 ± 0.061 | 0.003 ± 0.002 | 39 | 35 | 0.993 ± 0.0080 | 0.004 ± 0.003 |

Note:
Abbreviations are as follows: $N$, sample size; nh, number of haplotype; $h$, haplotype diversity, and $\pi$, nucleotide diversity.

phylogeographic framework to test whether the combination of these two approaches can effectively distinguish recently diverged lineages.

## MATERIALS AND METHODS

### Sample collection and DNA extraction

We collected 252 *C. viridis* between 2007 and 2017 from 15 locations across its Indo-Pacific distribution (Table 1; Fig. 1). We collected specimens by hand net and clove oil, either by scuba diving or snorkeling and preserved tissue samples (fin clips, a piece of muscle, or both) in 95% alcohol and stored at 4 °C. Large fish were released following fin clipping, but individuals too small for fin clipping were euthanized with clove oil and preserved whole in 95% ethanol. The sampling depth of all specimens used in present study was less than 10 m. This experiment doesn't involve animal experiment and the field sampling process complied with the regulation drafted by the Animal Care and Use Committee of National Sun Yat-sen University. Additionally, the permission to perform research activities in Indonesia was issued by the Indonesian Government and the Ministry of Research and Technology under research permit No. 272/SIP/FRP/SM/VII/2013.

We extracted genomic DNA from tissue fragments using Geneaid Tissue Genomic DNA mini Kit (Geneaid Biotech, New Taipei City, Taiwan) following manufacturers' protocol and eluted extracted DNA in TE buffer and stored at −20 °C until amplification by polymerase chain reaction (PCR).

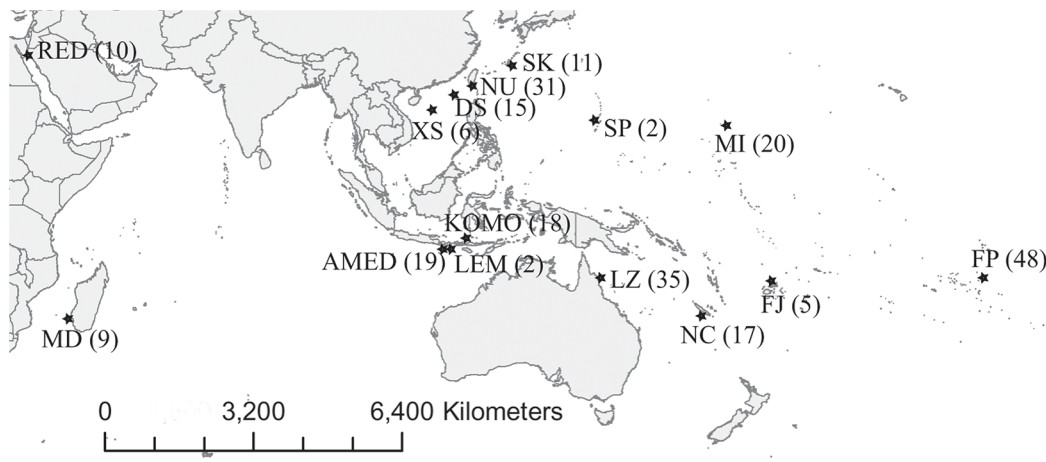

**Figure 1 Map of sampling locations.** Abbreviations of the locations are given in Table 1, and the number in the parenthesis is the sample size. 

## Amplification of genetic markers

We amplified a portion of the mitochondrial cytochrome b (Cytb) gene using universal primers GluDG-L and H16460 (*Palumbi, 1996*). PCRs reactions were 30 μL in volume, containing 10–40 ng template DNA, three μL 10X buffer, 0.2 mM dNTPs, 1.5 mM $MgCl_2$, 10 mM each primer, and 0.2 units of Taq polymerase (MDbio, Taipei). The thermocycling profile consisted of initial denaturation at 94 °C for 2 min, followed by 41 cycles of denaturation at 94 °C for 30 s, annealing at 57 °C for 30 s, and extension at 72 °C for 40 s, concluding with a final extension at 72 °C for 2 min.

Because mitochondrial and nuclear genes can have very different histories, we also amplified the nuclear recombination-activating (Rag2) gene for a subset of samples that represented distinct clades recovered in the mtDNA, using primers RAG2F and RAG2R (*Westneat & Alfaro, 2005*). PCR reactions were as above, except that we used a five mM MgCl concentration and the following thermocycling parameters: 39 cycles of denaturation at 94 °C for 30 s, annealing at 56 °C for 30 s, and extension at 72 °C for 40 s, and a final extension at 72 °C for 2 min.

The nucleotide sequences of the PCR products of both loci were determined using ABI 3730XL automated sequencer (Carlsbad, CA, USA) by Genomics (https://www.genomics.com.tw). Nucleotide sequences were assembled and edited using the SEQUENCER version 4.2 software (Gene Code, Ann Arbor, MI, USA). Sequences were uploaded to GenBank under the accession number MH743228–MH743691.

## Phylogenetic analyses

Prior to any analyses, we aligned DNA sequences from each gene region in Clustal W (*Thompson, Higgins & Gibson, 1994*) and exported these sequences to MEGA 6 (*Tamura et al., 2013*) to visually inspect all alignments for accuracy. To quantify genetic diversity measures, we calculated standard genetic diversity indexes including haplotype diversity (*h*) and nucleotide diversity (π) in Arlequin 3.5 (*Excoffier & Lischer, 2010*). We then inferred phylogenetic trees from each individual locus using maximum likelihood (ML)

and Bayesian inference performed on the CIPRES Science Gateway (*Miller, Pfeiffer & Schwartz, 2010*), and in BEAST 2.4.5 (*Bouckaert et al., 2014*). We conducted the ML analyses in RAxML version 8.1.24 (*Stamatakis, 2014*) using the GTR+G substitution model which selected by MEGA 6 (*Tamura et al., 2013*) as best-fit substitution model. For Bayesian inference, we used MrBayes version 3.2.2 (*Ronquist et al., 2012*), implementing two parallel runs of four simultaneous Markov chains for 10 million generations, sampling every 1,000 generations and using the default parameters. Run parameters employed unlinked substitution models, rated heterogeneity models, and based frequencies across partitions. In the Bayesian analyses the first one million generations (10%) were discarded as burn-in, based on log-likelihood tree scores. Meanwhile, the convergence diagnostic was applied and the stop probability was set to 0.01. Nodal support was evaluated individually for all trees using non-parametric bootstrapping with 1,000 ML replicates as employed in RAxML (*Stamatakis, 2014*), and by calculation of posterior probabilities as employed in MrBayes. Lastly, we generated a median-joining haplotype networks based on Cytb and Rag2 sequence datasets by using Popart 1.7 (http://popart.otago.ac.nz).

To date the ages of *C. viridis* clades, we arbitrarily subsampled 2 Cytb sequences from each clade; we also included two outgroups, including *C. atrilobata* (AY208524.1) and *C. multilineata* (AY208533.1) to calibrate date ranges as these two taxa diverged 3.1 Mya, (*Quenouille, Bermingham & Planes, 2004*). We created the XML BEAST input file using the software BEAUti v.1.8.2. (as implemented in BEAST), with the setting of 50 million generations under the uncorrelated relaxed clock model, and sampling trees once every 1,000 generations. To test for inter-run variation, we conducted two independent runs in BEAST 2.4.5 (*Bouckaert et al., 2014*), and then checked these runs for convergence with the software Tracer v1.5 (available at http://beast.bio.ed.ac.uk/Tracer/). After discarding the 20% burn-in, we pooled the remaining tree samples from the two runs into a combined file to calculate the maximum clade credibility tree. From this tree, we calculated the posterior mean divergence ages and 95% credibility intervals for all nodes using Tree Annotator v1.8.2 (implemented in the BEAST package). To compare the similarity of two gene tree topologies, Phylo.io software (*Robinson, Dylus & Dessimoz, 2016*) was used.

## Ecological niche characterization and comparison

To characterize the ecological niches of *C. viridis* (*C. viridis* Clade A) and potential cryptic species (*C. viridis* Clade B), we obtained geophysical, biotic, and environmental data layers for global sea surface from Bio-ORACLE (downloaded on February 20th, 2018; *Tyberghein et al., 2012*; *Assis et al., 2018*). We then extracted the values of those factors for the locations where *C. viridis* samples were collected. We excluded factors such as ice thickness and sea ice concentration from our analyses because all sample sites were tropical and ice free.

To explore niche differentiation between *C. viridis* Clade A and *C. viridis* Clade B, we first compared the values of individual environmental factors between *C. viridis* Clade A and Clade B sampling sites using Mann–Whitney tests. Next, we plotted niche
differences in a two-dimensional non-metric multidimensional scaling (NMDS) space based on the Euclidean dissimilarity of those factors. Due to the different units of the environmental factors, we standardized factor values before running the NMDS analysis. Lastly, we tested equivalency of ecological niches between the species based on environmental factors of *C. viridis* Clade A and Clades B sampling locations in a principle components analysis (PCA) framework. Due to potential correlations among environmental factors, we ran the PCA on all environmental factors and then used the resulting principal components to define niches of the species. Because *C. viridis* mainly occur in coral reefs, we restricted our PCA to regions (grid cells) where the maximum depth was equal or less than 50 m based on the bathymetry data obtained from MARSPEC (http://www.marspec.org/; *Sbrocco & Barber, 2013*). To make the spatial resolution of the bathymetry data layer, which is originally at 30-by-30-arc second resolution, consistent with the Bio-ORACLE data, we upscaled it to 5-by-5-arc min resolution with each pixel value in the upscaled data layer being the maximum value over $10 \times 10$ original pixels within that new pixel.

To measure niche similarity between *C. viridis* Clade A Clade B, we calculated the *D* and *I* statistics developed by *Warren, Glor & Turelli (2008)*. These two indices range from zero and one with higher values indicating more similar niches. To test for niche equivalency between these two cryptic lineages of *C. viridis*, we compared the observed similarity values to null distributions of similarity obtained through a randomization procedure under a hypothesis of an identical niche (*Warren, Glor & Turelli, 2008*), using the "dismo" package in R (version 3.4.2).

## RESULTS

We sequenced 911 bp of mitochondrial Cytb, and 743 bp of nuclear gene (Rag2, 743 bp) from 248 individual *C. viridis*, representing 15 locations across the Indo-Pacific (Fig. 1). In total, there were 135 unique haplotypes of Cytb; nucleotide diversity ($\pi$) ranged from 0.0033 to 0.0544, and haplotype diversity (*h*) ranged from 0.8824 to 1 among locations. For Rag2, 30 samples would not amplify, resulting in a total of 218 Rag2 sequences, representing fourteen unique haplotypes after omitting those sites with ambiguity codes (e.g., Y and K), additionally, there were 40 parsimony informative sites after alignment and 26 (heterozygotes) out of 40 were contained ambiguous signals. The nucleotide diversity ($\pi$) and haplotype diversity (*h*) of Rag2 ranged from 0 to 0.0154 and 0.9933 to 1, respectively, (Table 1). These two genetic diversity indexes could be over-estimated since those ambiguous sites were been considered as variable sites during computation in Arlequin 3.5.

The ML tree based on Cytb gene revealed three deeply divergent lineages within *C. viridis*; Clade A, Clade A-1, and Clade B (Fig. 2). In contrast, the corresponding Rag2 likelihood tree only differentiated Clades A and B, with no additional divergence within Clade 1. However, Clades A and B were not concordant across Cytb and Rag2; four mtDNA Clade B samples, one from Komodo and three from Lizard Island, clustered with Clade A based on Rag2. Additionally, one sample from Komodo contained a Clade B Rag2 sequence, but a Clade A Cytb sequence (Table S1).

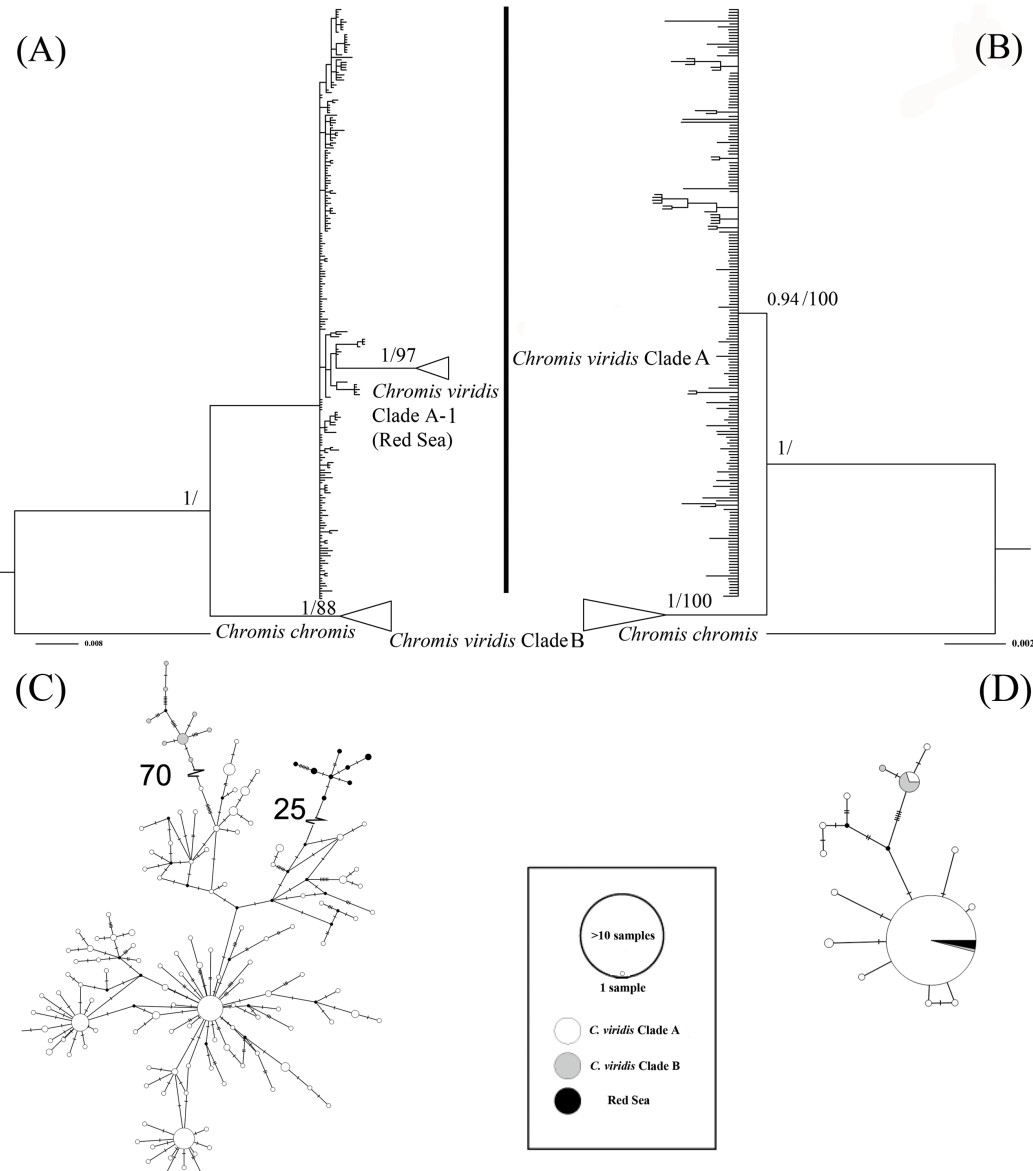

**Figure 2 Bayesian phylogenetic tree (A, B) and corresponding haplotype network (C, D) based on two genetic markers including *Cytb* and Rag2.** Nodes are presented only for those with bootstrap scores >85% majority rule for maximum likelihood and >90% majority probabilities for Bayesian probability values (BI/ML). For the Haplotype network, different colors indicate different clades (e.g., white = *Chromis viridis* Clade A, gray = *Chromis viridis* Clade B, and black = Red Sea), mutation step larger than 20 were denoted by hatch marks with number of mutation steps.

Whether based on Cytb or Rag2, all Clade B haplotypes came from only three locations; Lizard Island on the Great Barrier Reef, Komodo Island in Indonesia, and the island of Fiji. In contrast, Clade A occurred broadly throughout the Indo-Pacific, including all three Clade B localities. Based solely on mtDNA, Clade A was further divided into two clades; Clade A was broadly distributed, and Clade A-1 was restricted to the Red Sea, although the Red Sea clade was not recovered in the Rag2 phylogeny. The general tree topology

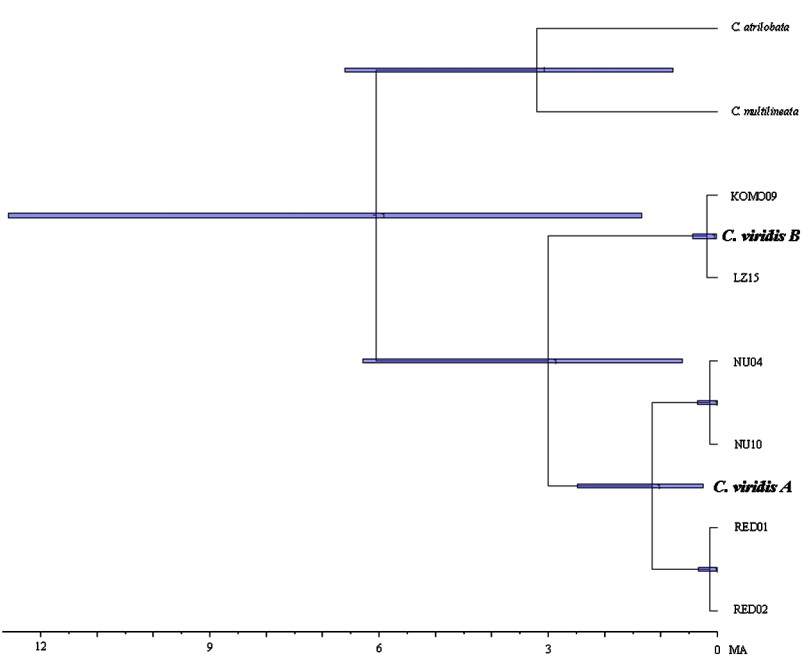

**Figure 3 Time-tree of cryptic lineages among *Chromis viridis* obtained from BEAST.** With a 3.1 Mya time constraint on the node between *Chromis atrilobata* and *Chromis multilineata* (*Quenouille, Bermingham & Planes, 2004*). Horizontal gray bars at nodes indicate 95% posterior probability densities (HPD) intervals of age. 

of these two genes was highly similar (Fig. S1), except the Clade A-1 and those samples clustered with it, which can be revealed only by Cytb.

The patterns from the median joining network tree were identical to the phylogenetic trees based on either Cytb and Rag2. However, a few individuals assigned to Clade B based on Cytb clustered with Clade A based on their Rag2 sequences (Fig. 2).

Results from BEAST indicated that *C. viridis* Clade A and Clade B diverged approximately 3 Mya (95% HPD: 0.631–6.291 Mya), with the divergence between Clade A and A-1 dating to 1.165 Mya (95% HPD: 0.247–2.49 Mya), and crown age of A-1 (Red Sea) was dated 0.138 Mya (95% HPD: 0.013–0.332 Mya) (Fig. 3).

## Niche divergence tests

Results spanning 11 environmental factors (Table 2) showed no significant difference in ecological niche parameters of Clade A or Clade B of *C. viridis*. Moreover, there was also no significant environmental difference between the locations where two clades co-occur and those where only Clade B occurs (Table 2), suggesting that the environmental factors cannot explain why Clade B co-occurs with Clade A at some locations, but not at others.

The NMDS analysis also showed a lack of niche differentiation. The first two NMDS axes explained 86.7% of the variation in the 11 environmental factors across the 15 sampling sites. The three sampling sites where *C. viridis* B occurs fall at the center of the convex hull of all sampling sites in the two-dimensional NMDS space (Fig. 4), indicating complete niche overlap between Clades A and B. The first four principal components of the environmental factors, which were standardized before the analysis, had eigenvalues higher

**Table 2 Mann–Whitney *U*-tests for the environmental differences between presence locations of clade A and those of clade B, and between presence and absence locations of clade A.**

| Environmental factor | A presence (Mean ± SD) | B presence (Mean ± SD) | A absence (Mean ± SD) | A presence vs. B presence | | A presence vs. A absence | |
|---|---|---|---|---|---|---|---|
| | | | | *U* | *P*-value | *U* | *P*-value |
| Temperature (°C) | 27.44 ± 0.86 | 27.15 ± 1.35 | 27.08 ± 1.47 | 20.5 | 0.86 | 20 | 0.84 |
| Salinity (PSS) | 34.31 ± 0.86 | 34.55 ± 1.43 | 34.61 ± 1.57 | 22.5 | 1 | 18 | 1 |
| Current velocity (m$^{-1}$) | 0.09 ± 0.06 | 0.13 ± 0.19 | 0.14 ± 0.21 | 23.5 | 0.95 | 17 | 0.95 |
| Nitrate (mol · m$^{-3}$) | 0.007 ± 0.009 | 0.11 ± 0.31 | 0.13 ± 0.34 | 25.5 | 0.77 | 15 | 0.73 |
| Phosphate (mol · m$^{-3}$) | 0.25 ± 0.05 | 0.23 ± 0.06 | 0.22 ± 0.06 | 18.5 | 0.68 | 22 | 0.63 |
| Silicate (mol · m$^{-3}$) | 3.57 ± 1.99 | 3.74 ± 1.85 | 3.79 ± 1.90 | 23.5 | 0.95 | 17 | 0.95 |
| Dissolved molecular oxygen (mol · m$^{-3}$) | 203.15 ± 1.50 | 203.65 ± 3.34 | 203.78 ± 3.70 | 22.5 | 1 | 18 | 1 |
| Iron (µmol · m$^{-3}$) | 0.001 ± 0.0003 | 0.0007 ± 0.0005 | 0.0006 ± 0.0005 | 10.5 | 0.17 | 30 | 0.1 |
| Chlorophyll (mg · m$^{-3}$) | 0.14 ± 0.04 | 0.15 ± 0.12 | 0.15 ± 0.13 | 18.5 | 0.68 | 22 | 0.63 |
| Phytoplankton (µmol · m$^{-3}$) | 1.19 ± 0.16 | 1.14 ± 0.48 | 1.13 ± 0.54 | 15.5 | 0.44 | 25 | 0.36 |
| Primary productivity (g · m$^{-3}$ · day$^{-1}$) | 0.006 ± 0.003 | 0.006 ± 0.008 | 0.007 ± 0.009 | 17.5 | 0.59 | 23 | 0.54 |

than 1. Together they explain about 83.1% of the total variation across all the grid cells with the maximum water depth less than or equal to 50 m. Based on the component scores corresponding to the locations where Clades A and B were found, niche similarity indices were high, with $D = 0.841$ and $I = 0.973$. Moreover, niche equivalency tests showed that none of the two values were significantly different from the values obtained through a randomization process under the hypothesis of an identical niche (*P*-values for $D$ and $I$ were 0.311 and 0.294, respectively).

## DISCUSSION

Range-wide phylogeographic analyses of the blue green damselfish, *C. viridis* revealed two divergent lineages in both mitochondrial Cytb and nuclear Rag2 DNA sequence data. These cryptic lineages were first reported by *Froukh & Kochzius (2008)* in the Coral Triangle and Red Sea, and subsequently by *Messmer et al. (2012)* who examined Australia and French Polynesia. However, these studies each only covered a fraction of the geographic range of *C. viridis*. By examining patterns across its entire range, this study shows that these cryptic lineages of *C. viridis* are sympatric over a portion of their Pacific range.

Moreover, previous studies (*Froukh & Kochzius, 2008*; *Messmer et al., 2012*) only used mtDNA markers, providing incomplete insights into genetic structure due to its maternal inheritance (*Prugnolle & De Meeûs, 2002*; *Daly-Engel et al., 2012*). By sequencing both mtDNA and nuclear markers, this study confirms the presence of two cryptic clades in *C. viridis*, Clades A and B. These clades diverged approximately 3 Mya similar to the divergence of *C. atrilobata* and *C. multilineata* that were separated by Isthmus of Panama (*Domingues et al., 2005*). Given the depth of this divergence and the largely concordant
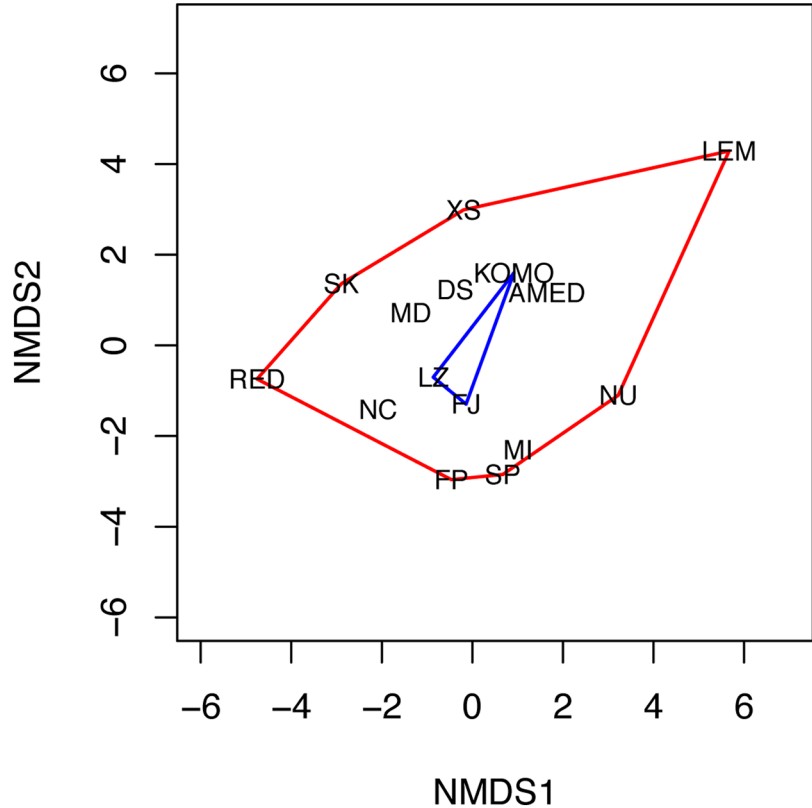

**Figure 4 Non-metric multidimensional scaling (NMDS) plot of the 15 sampling sites.** The two NMDS axes (NMDS1 and NMDS2) represent environmental gradients defined by the 11 environmental factors examined across the sites. The convex hulls for the sampling sites where *C. viridis* A and *C. viridis* B were found are shown in red and blue, respectively. Please see the legend of Fig. 1 for the abbreviation of the site names.                                     

differentiation of Clade A and B in Cytb and Rag2, it is likely that these two lineages represent distinct species, raising interesting questions about the sympatry of Clades A and B in part of their range.

## Origin of lineage diversification

Although the divergence of Clades A and B of *C. viridis* is clear, the origin of this divergence is not. Within the Indo-Pacific region, the "Indo-Pacific Barrier" has been considered as a soft barrier that could be the main driving force of marine biological provinces in this region (*Gaither et al., 2011*). *Barber & Bellwood (2005)* and *Cowman & Bellwood (2013)* examined the importance of this barrier by evaluating the extent of temporal concordance in vicariance in three prominent families of reef fish, including Labridae, Pomacentridae, and Chaetodontidae. Both studies showed that the isolation effect of Indo-Pacific barrier on the widely distributed fishes mainly occurred mostly between end Miocene and Early Pliocene (2–6 Myr), and the majority of vicariance events occurred in a narrow time interval at approximately 2.5 Myr.

The date of divergence of Clade A and B broadly conforms to the onset of Plio-Pleisetocene sea level fluctuations (*Voris, 2000*). However, Clade B populations of *C. viridis* occur

on islands surround by deep water where lowered sea levels would not result in land barriers that promote vicariance, unlike other marine taxa distributed on opposite sides of the Sunda Shelf (*Barber, Erdmann & Palumbi, 2006*; *Crandall et al., 2008*, *2014*; *DeBoer et al., 2014a*, *2014b*; *Waldrop et al., 2016*; *Simmonds et al., 2018*). Given the Indo-Pacific wide range of *C. viridis* and a divergence time between two cryptic clades dating to 3 Mya, isolation across the Indo-Pacific Barrier is the most likely driver of divergence between Clades A and B. Under this scenario, Clade A would be an Indian Ocean clade that has expanded into the Pacific, where it now overlaps with the Pacific Clade B, a process previously noted in Neritid snails (*Crandall et al., 2008*).

An alternate, but not mutually exclusive explanation for divergence of Clade B comes from recent studies of coral associated marine taxa. Similar to phytophagus insects that undergo ecological divergence associated with host switching (*Berlocher & Feder, 2002*; *Hébert, Scheffer & Hawthorne, 2016*), recent studies from the marine realm demonstrate that changes in coral host taxa can promote lineage divergence, potentially leading to speciation (*Simmonds et al., 2018*). Samples were not collected in a way that allows us to test this hypothesis, but future studies separating samples by coral host to determine whether individuals from Clades A and B exist in mixed schools in sympatric populations, and if so, whether those schools are associated with different coral hosts.

The geographic distribution of Clade B is curious in that it is observed in Komodo, but not in other populations in the Lesser Sunda Islands (e.g., Amed, Nusa Lembongan). Similarly, Clade B is observed on the Great Barrier Reef and Fiji, but not in New Caledonia, a population located between these too population. One explanation for this disjunct distribution is that Clade B individuals are relatively rare, and that greater sampling intensity would reveal Clade B haplotypes in adjacent ranges, as would be expected. Alternatively, it is possible that Clade A is gradually displacing Clade B populations, and the areas of sympatry represent locations where this process is incomplete. Similar arguments were made to explain sympatry of highly divergent clades of marine snails in the Pacific Ocean (*Crandall et al., 2008*).

In contrast to the divergence of Clade A and B, the divergence of Clade A-1, unique to the Red Sea, is more easily interpreted as the result of vicariance. The Red Sea is a semi-enclosed basin that is frequently invoked in driving population differentiation of reef organisms between the Red Sea and Indian Ocean (*DiBattista et al., 2013*). Additionally, the 95% HPD of Red Sea Clade is ranged from 0.013 to 0.332 Mya which aligns with the most recent closure of the Red Sea (*Siddall et al., 2003*). However, while mitochondrial DNA shows a well-supported Red Sea sub clade within *C. viridis* Clade A, this clade is not confirmed with nuclear Rag2 sequence.

Lack of concordance between the mtDNA and nuclear phylogenies is not surprising. First, there is substantially more genetic variation in the mtDNA sequences because DNA repair mechanisms are less efficient in mtDNA, resulting in much higher substitution rates (*Alexeyev et al., 2013*). Second, because mitochondrial genome is maternally inherited and haploid, its effective population size is one-fourth that of a nuclear gene, meaning that lineage sorting occurs more rapidly in mtDNA (*Hare, 2001*). Thus, mitochondrial gene trees have a substantially higher probability of accurately recovering recently divergence

lineages with short internode distances than do nuclear genes (*Moore, 1995*). Given that the Red Sea mtDNA clade only dates to 0.138 Mya, it is unsurprising that divergence in this region is not recovered in the Rag2 sequences. Thus, while the Red Sea population is clearly a sub-population of *C. viridis* Clade A, it is unlikely a cryptic species, like Clades A and B.

## Cryptic species

Many phylogeographic studies in the Indo-Pacific find highly divergent clades, often with regions of sympatry (*Barber et al., 2002*; *Barber, Erdmann & Palumbi, 2006*; *Crandall et al., 2008*; *Gaither et al., 2011*; *Liu et al., 2012*; *DeBoer et al., 2014a*; *Bowen, Karl & Pfeiler, 2007*). In many cases, these cryptic clades may represent cryptic species, similar to *Liu et al. (2012)* where a divergent clade of *Pomacentrus coelestis* in Micronesia was subsequently described as a new species (*Liu, Ho & Dai, 2013*). The concordant phylogeographic patterns in mtDNA and nuclear DNA, and the depth of this divergence suggests that Clades A and B of *C. viridis* may represent two cryptic species with different geographic ranges; Clade A is a widely distributed Indo-Pacific, while Clade B is only found in Lizard Island, Komodo, and Fiji.

Comparison of Cytb and Rag2 trees revealed a limited amount of discordance. Given the relatively recent divergence, one potential explanation for this pattern is incomplete lineage sorting (*Tang et al., 2012*). Incomplete lineage sorting is the simplest explanation for individuals with Clade B *C. viridis* haplotypes in the mtDNA tree having Clade A *C. viridis* Rag2 sequences (Table S1). However, incomplete lineage sorting could not explain the sequence of komo4 clustered with *C. viridis* clade B in the Rag2 tree but clustered with *C. viridis* Clade A in the mtDNA tree. The later finding could likely explain by the consequence of hybridization.

Hybridization is in *C. viridis* is unsurprising as closely related Pomacentrids often have overlapping geographic distributions, co-occur in the same microhabitats (e.g., colonies of branching corals) (*Randall & Allen, 1977*). Hybridization has been observed in several sibling species of damselfish, including *Abudefduf abdominalis* × *Abudefduf vaigiensis* (*Coleman et al., 2014*), *Amphiprion chrysopterus* × *Amphiprion sandaracinos* (*Gainsford, Van Herwerden & Jones, 2015*), *Amphiprion mccullochi* × *Amphiprion akindynos* (*Van Der Meer et al., 2012*), *Dascyllus carneus* × *D. marginatus* (*DiBattista et al., 2015*), and *D. reticulatus* × *D. aruanus* (*He et al., 2017*). As such, both hybridization and incomplete lineage sorting are likely responsible for the discordant patterns in the two markers.

## Incongruence between genetic and ecological niche divergence

*Hutchinson (1978)* proposed that two species cannot occupy the same ecological niche, yet niche conservatism predicts that closely related taxa retain ancestral ecological affiliations and persists in similar environments (*Lord, Westoby & Leishman, 1995*; *Webb et al., 2002*; *Wiens & Graham, 2005*). Allopatric sister taxa are often characterized by niche conservatism, but because geographic isolation drives speciation (*Peterson, Soberón & Sánchez-Cordero, 1999*; *Peterson et al., 2001*; *Kozak & Wiens, 2006*) sibling species can't

compete for the same niche space, because, by definition, they have non-overlapping geographic ranges.

The alternative to niche conservatism is niche divergence, in which sister taxa occupy different niches (*Dayan & Simberloff, 2005*). Niche divergence is typically associated with sympatric speciation as diversification results from reduction in gene flow associated with divergence in traits with ecological function (e.g., habitat segregation, pollinator divergence, behavioral changes, phenological shifts, and mating system shifts). Under niche divergence, young sister species with high degrees of range overlap should ecologically diverge (*Dayan & Simberloff, 2005*; *Davies et al., 2007*). While ecological modeling and phylogenetics are commonly integrated to understand the relationship between evolutionary and ecological divergence of sibling species (*Kalkvik et al., 2012*; *Schorr et al., 2012*), few studies examine the intra-specific (cryptic lineage) level (*Gutiérrez-Tapia & Palma, 2016*).

Our results show that Clade A and B of *C. viridis* are sympatric, with Clade B haplotypes occurring only at three localities. There are two possible explanations for this pattern. First, lineage diversification could have occurred in sympatry, as seen in Indo-Pacific coralivorous snails (*Simmonds et al., 2018*). Alternatively, diversification could have occurred in allopatry with secondary overlap in geographic ranges as proposed for neritid snails (*Crandall et al., 2008*). Niche comparisons based on 11 environmental factors showed that Clade A has a broader ecological niche than Clade B does (Fig. 4). Despite this difference, there is no significant niche differentiation between these two lineages. As such, diversification is more likely to have arisen with niche conservatism rather than niche differentiation.

Given the absence of niche differentiation, our results suggest that the diversification of Clades A and B most likely occurred in allopatry, and that secondary contact is the most likely explanation for their current distribution. If true, niche conservatism could help explain the disjunct distribution of Clade B. Under this scenario, Clade A and Clade B would compete for the same niche space when they occur in sympatry. If Clade A is a superior competitor, it would gradually displace Clade B populations, potentially explaining the relative rarity of Clade B haptotypes and their disjunct distributions.

Alternatively, *Pyron et al. (2015)* suggested that niche conservatism could result from soft allopatry where there is low environmental heterogeneity. As noted above, the shallow Sunda and Sahul continental shelves exposed during low sea level stands (*Voris, 2000*), forming long land bridges that restricted larval exchange between the tropical Indian Ocean and the western Pacific (reviewed in *Randall, 1998*). However, because deep water passages in what is modern day Indonesia remained open, and because low sea level stands were intermittent, the Indo-Pacific Barrier has been considered a soft dispersal barrier for marine taxa (*Cowman & Bellwood, 2013*). The 11 environmental factors we used to define the niches of the two lineages of *C. viridis* only reflect broad-scale environmental conditions. However, there could be subtle variation in environmental factors, or variation could occur at finer scales (e.g., microhabitat), resulting in soft-allopatric divergence.

It is important to note that the broad environmental variation we examine does not capture the potential diet shifts observed in sibling species (*Goodheart et al., 2017*),

novel traits (*Liu et al., 2018*), and/or micro habitat preference (*Whitney, Donahue & Karl, 2018*) that could act to drive or reinforce lineage diversification. The latter is particularly intriguing given recent studies demonstrating ecological speciation resulting from shifts in coral hosts (*Simmonds et al., 2018*). Further studies such as stable isotope analysis and detailed morphological examinations in a phylogenetic framework are needed to better understand the ecological and morphological divergence between these two lineages.

## CONCLUSIONS

Although cryptic diversification in widespread marine species is common (*Hubert et al., 2012*), phylogeographic studies typically ignore the potential role of ecological niche partitioning on lineage diversification. The present study shows evolutionary divergence between two lineages of *C. viridis* that have overlapping ecological niches, supporting niche conservatism. It is unclear whether this pattern results from allopatric divergence with secondary contact, or from subtle differentiation in ecological niches not captured by the broad scale environmental data used to compare ecological niches. Regardless of the origins, the substantial genetic divergence between the two clades of *C. viridis* in both mtDNA and nuclear loci strong suggest that they are cryptic taxa. Because *C. viridis* is highly exploited in the aquarium trade (*Wabnitz, 2003*), we suggest that a higher conservation priority needs to be given on the Clade B lineages restricted to Australia, Indonesia, and Fiji to protect this unique genetic lineage.

## ACKNOWLEDGEMENTS

We thank Dr. S. Cheng, Dr. M-J Ho, Dr. Y-R Cheng, F-T Chang, N. Narendra and A. Sembiring for their assistant in the field, and Dr. V. Messmer, Dr. B. Frédérich, Dr. F. Borsa, Professor S. Planes and S. Johnson for sharing tissue samples. Special thanks to A. C. Bentley, for his curatorial help on obtaining tissue samples from Ichthyology session of Kansas University.

### Funding

This study was funded by Ministry of Science and Technology, Taiwan (Grant Number: MOST 106-2611-M-110-009). The funders had no role in study design, data collection and analysis, decision to publish, or preparation of the manuscript.

### Grant Disclosures

The following grant information was disclosed by the authors:
Ministry of Science and Technology, Taiwan: MOST 106-2611-M-110-009.

### Competing Interests

The authors declare that they have no competing interests.

## Author Contributions

- Shang Yin Vanson Liu conceived and designed the experiments, performed the experiments, analyzed the data, contributed reagents/materials/analysis tools, prepared figures and/or tables, authored or reviewed drafts of the paper, approved the final draft.
- Mao-Ning Tuanmu conceived and designed the experiments, performed the experiments, analyzed the data, contributed reagents/materials/analysis tools, prepared figures and/or tables, authored or reviewed drafts of the paper, approved the final draft.
- Rita Rachmawati performed the experiments, contributed reagents/materials/analysis tools, approved the final draft.
- Gusti Ngurah Mahardika contributed reagents/materials/analysis tools, approved the final draft.
- Paul H. Barber contributed reagents/materials/analysis tools, authored or reviewed drafts of the paper, approved the final draft.

## Animal Ethics

The following information was supplied relating to ethical approvals (i.e., approving body and any reference numbers):

The field sampling process was approved by the Animal Care and Use Committee of National Sun-Yat-Sen University.

## Field Study Permissions

The following information was supplied relating to field study approvals (i.e., approving body and any reference numbers):

The sampling in Indonesia was under research permit No. 272/SIP/FRP/SM/VII/2013 issued by the Indonesian Government and the Ministry of Research and Technology.

## Data Availability

Sequences are available at GenBank under the accession number MH743228–MH743691.

## Supplemental Information

Supplemental information for this article can be found online at http://dx.doi.org/10.7717/peerj.7384#supplemental-information.

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
