# Peer review of "Integrating phylogeographic and ecological niche approaches to delimitating cryptic lineages in the blue–green damselfish (Chromis viridis)"

_PeerJ, doi:10.7717/peerj.7384_

## Round 0.1 · original submission · Minor Revisions

Overall both referees are quite positive about the paper and provide relatively simple feedback for minor revisions to improve the text. I do not expect the suggested changes will be difficult for you to address, and I look forward to seeing your revised manuscript.

Reviewer 1 ·

Basic reporting

Overall the paper is well-written, the language and structure of the paper are easy to follow and the cited literature is sufficient and appropriate for the scope of this study. I did find the figures a bit blurry and font sizes difficult to read; I encourage the authors to re-craft the figures with larger font and higher resolution lines, especially the map (Figure 1).

Experimental design

The examination of possible cryptic species in the marine environment using direct sequencing has become fairly common and the methods used are well-vetted. Again, the originality of this paper to past studies is the integration of ecological niche modeling to test phylogenetic patterns. Nicely done.

Validity of the findings

The methods were provided with sufficient detail to replicate. Greater sample size would have been ideal for several of the locations (Siapan only had 2 fish), but I can appreciate the difficulty of obtaining samples from such a broad geographic range. The study’s overall sample size of several hundred provides enough power to give confidence to the drawn conclusions.

Additional comments

Thank you for this opportunity to review for PeerJ. I have read the manuscript ‘Integrating phylogeographic and ecological niche approaches to delimitating cryptic lineages in the blue-green damselfish (Chromis viridis)’. This paper takes a complementary approach to examining the population-level/cryptic species-level relationship of a widely distributed coral-reef associated marine fish using a clever combination of molecular genetic data and ecological niche data. I commend the authors for attempting this analysis. Although the ecological niche data did little to explain the observed genetic divergence among the C. viridis clades in this study, I believe it provides a replicable road map for others to integrate environmental and genetic data to explain marine biodiversity. The paper already inspires re-thinking of how my lab will look at including ecological niche data in our pop gen work.

Overall the paper is well-written, the language and structure of the paper are easy to follow and the cited literature is sufficient and appropriate for the scope of this study. I did find the figures a bit blurry and font sizes difficult to read; I encourage the authors to re-craft the figures with larger font and higher resolution lines, especially the map (Figure 1).

The examination of possible cryptic species in the marine environment using direct sequencing has become fairly common and the methods used are well-vetted. Again, the originality of this paper to past studies is the integration of ecological niche modeling to test phylogenetic patterns. Nicely done. The methods were provided with sufficient detail to replicate. Greater sample size would have been ideal for several of the locations (Siapan only had 2 fish), but I can appreciate the difficulty of obtaining samples from such a broad geographic range. The study’s overall sample size of several hundred provides enough power to give confidence to the drawn conclusions.

I have several minor comments for the paper below. I believe the paper should be approved for publication after considering the below comments, and those of the other reviewers.

Comments:
Line 73-76 – The authors cite possible cryptic lineages of C. viridis based on mtDNA data, and earlier mentioned how cryptic species are sometimes distinguished using morphology (Line 54). Are there examples where morphological data (or possibly other types of data) gives an indication of cryptic lineages in C. viridis? That is data other then the mtDNA already mentioned?

Line 99 and Line 108: The authors amplify Cytb and Rag2 primers, yet referenced papers of Froukh & Kochzius used CR-A/XAN-DL-R primers and Messmer et al 2012 used CR-A/CR-E primers. The Control Region (CR) arguably evolves faster than other parts of the mtDNA like Cytb (Avise et al. 1987). What is the authors justification for using Cytb and even Rag2 over CR which past authors have used and may provide a comparison with these other data sets? Furthermore, the authors themselves highlight differences in substitution rates of the genetic markers used (Line 318-325). Again, why not include CR?

Line 242-244 – The authors bring up the role of water depth in helping to explain total variance in the PCA, however, I did not see that depth where fish were captured was included in the method. Do I read this correctly that all C. viridis occur at 50m depth or less, and that all samples were obtained at these depths (seems likely given stated sampling method of SCUBA/snorkeling), and thus no further exploration was done of the data to tease apart role of depth in differentiating the clades? The discussion on host switching leading to speciation (Line 294), although not testable due to the sampling scheme, further makes me wonder if environmental factors such as depth may lend themselves to explaining the divergence between the clades.

Figure 1 – Map is somewhat of low quality. Can this be improved upon? Further, I would suggest zooming in closer (cutting out the far north and south latitudes) to allow easier distinction of sites in Indonesia.

Reviewer 2 ·

Basic reporting

The paper is very well written and the integration of phylogeography and ecology would make it of interest to a broad audience. The literature references are comprehensive and relevant. The quality, format and arrangement of figures are all more than adequate to summarize their findings and support their conclusions.

I added some minor suggestions for improved clarity.

Experimental design

The experimental design is sound. Overall, this is a good study that address the role of niche divergence in the context of species divergence. Niche divergence is often stated as a mode for speciation in phylogeographic studies, but rarely ever tested in parallel. I suggested as a slight addition to the BEAST results to explain the patterns seen with the Red Sea

Validity of the findings

The conclusions are well stated and the findings are valid and well supported. They linked very well to the
original research objective and placed in the context of previous research.

Additional comments

Lines: 128-129: How was this model determined? Please describe and add to the methods.

Line 227: Please list the 95% HPD ranges

Lines 312-313: What is the estimated age of the Red Sea population based on the BEAST analysis? It appears to be around 20,000 year which aligns well with the most recent closure of the Red Sea (Siddall et al., 2003)

Line 325: The 1.2 my age applies to Clade A not just to the Red Sea. The Red Sea appears to be much younger

Figure 2: change 0.9443 to 0.94 to keep consistent significant values

Figure 2, Haplotype networks: The network for cyt b is difficult to distinguish between the mutation tick marks and the circles that identify Red Sea haplotypes. There is room to blow up the size of the Rag2 network to make it easier to read

Figure S1: there is no caption. I am not sure what this figure represents

Suggested reference:
Siddall, M., Rohling, E.J., Almogi-Labin, A., Hemleben, C., Meischner, D., Schmelzer, I., Smeed, D.A., 2003. Sea-level fluctuations during the last glacial cycle. Nature 423, 853.

---

## Round 0.2 · accepted · Accept

Thank you for your detailed response to referee feedback. I believe it has made an already good manuscript that much stronger, and I am happy to accept this revised submission.